# The need for balanced dengue vaccine protection: Insights from Thai surveillance data on four serotypes

Ruchira R. Khosavanna[1,2], Sopon Iamsirithaworn[3], Soontorn Pinpaiboon[4], Kanyarat Phutthasophit[5], Taweewun Hunsawong[5], Albert I. Ko[6], Kathryn B. Anderson[2‡], Darunee Buddhari[5‡*]

1 Department of Medicine, Yale School of Medicine, New Haven, Connecticut, United States of America, 2 Microbiology and Immunology, State University of New York Upstate Medical University, Syracuse, New York, United States of America, 3 Thai Ministry of Public Health, Nonthaburi, Thailand, 4 Kamphaeng Phet Hospital, Kamphaeng Phet, Thailand, 5 Department of Virology, Walter Reed Army Institute of Research - Armed Forces Research Institute of Medical Sciences, Bangkok, Thailand, 6 Department of Epidemiology of Microbial Diseases, Yale School of Public Health, New Haven, Connecticut, United States of America

‡ Co-senior authors.
* daruneet.fsn@afrims.org

## Abstract

Dengue virus (DENV), comprising four distinct serotypes (DENV-1 to DENV-4), poses a major public health challenge in tropical regions. Infection with one serotype confers long-term immunity to that serotype alone, while subsequent heterologous infections are associated with increased risk of severe disease, necessitating vaccines that induce durable, balanced immunity across all serotypes. However, achieving such balance immunity remains a central challenge for dengue vaccine development. Using passive surveillance data from Kamphaeng Phet, Thailand (2004–2022), we characterized long-term serotype circulation and contributions to clinical dengue burden in a hyperendemic setting. We observed sustained co-circulation of all four serotypes over nearly two decades, with periodic shifts in dominant serotype. Among 6,840 PCR-confirmed dengue cases, the majority of which were hospitalized, DENV-1 through DENV-4 accounted for 32.8%, 25.9%, 24.8%, and 16.5% of detected dengue cases, respectively. Compared with DENV-1, clinically-apparent DENV-2 and DENV-4 infections were more likely to represent secondary infections (odds ratio 4.94 and 5.24, respectively) and occurred at older ages, underscoring the context-dependent clinical expression of different serotypes. Together, these findings demonstrate that all four dengue serotypes contribute meaningfully to clinical disease burden in Thailand and caution against de-emphasizing individual serotypes based on transient epidemiologic patterns. These results reinforce the importance of tetravalent vaccine strategies alongside continued evaluation of vaccine performance in evolving epidemiologic settings.

**Data availability statement:** The data that support the findings of this study are publicly available from Figshare with the identifier https://doi.org/10.6084/m9.figshare.31374625.

**Funding:** The original surveillance studies were supported in part by Military Infectious Diseases Research Program (MI230060) and a program project award from the United States National Institutes of Health (P01AI034533 and R01AI175941). RRK was supported by NIH/NIAID 5T32AI007517-23 awarded to Yale University Infectious Diseases Research Training Program. She received additional support from the Fogarty International Center of the National Institutes of Health under grant #D43TW009345 awarded to the Northern Pacific Global Health Fellows Program and grant #D43TW010540 awarded to the Global Health Emerging Scholars Program. The funders had no role in the study design, data collection and analysis, interpretation, decision to publish, or preparation of the manuscript. The contents are solely the responsibility of the authors and do not necessarily represent the official views of the National Institutes of Health.

**Competing interests:** The authors have declared that no competing interests exist.

## Author summary

Dengue is a mosquito-borne disease that infects millions of people every year. The virus consists of four serotypes. Infection with one serotype does not provide full protection against the others, and a second infection with a different serotype can increase the likelihood of severe illness. For this reason, understanding which serotypes are circulating is important for designing and evaluating effective vaccines. Our analysis of nearly two decades of surveillance data from Kamphaeng Phet, Thailand demonstrates that all four serotypes were consistently present over time, each contributing a substantial number of cases. The serotypes showed distinct age- and immunity-dependent epidemiologic patterns. No single serotype could be considered unimportant. These findings highlight the complex nature of DENV transmission in Thailand and emphasize the need for vaccines that provide protection against all four serotypes. Continuous monitoring of circulating serotypes is essential to guide vaccine development and to ensure their effectiveness in real-world settings.

## Introduction

Dengue is a mosquito-borne disease endemic to tropical and subtropical regions, placing billions of people at risk of infection. Dengue virus (DENV), a member of the Flaviviridae family and *Orthoflavivirus* genus, comprises four antigenically distinct serotypes (DENV-1 to DENV-4), each independently capable of causing disease ranging from asymptomatic infection to life-threatening manifestations. Global incidence of dengue has increased over recent decades; in 2024, reported cases exceeded 10 million, more than doubling the 4.6 million cases recorded in 2023, which itself represented a record year [1–3]. Climatic changes and rapid urbanization are key drivers of this rise, with warming temperatures expanding the habitat range of *Aedes* mosquitoes and facilitating viral transmission [4]. When dengue spreads into new geographical areas, outbreaks often result in high mortality during the initial phase [5]. Although supportive care with intravenous fluid management remains the cornerstone of treatment and has been shown to reduce mortality [6], these trends highlight the growing public health challenge posed by dengue and the urgent need for effective prevention strategies, particularly vaccination.

DENV infection with one serotype induces durable homotypic immunity but only transient protection against heterologous serotypes; subsequent heterologous infection is associated with increased risk for severe disease through antibody-dependent enhancement (ADE) [7]. Additional immune factors—including original antigenic sin, cell-mediated immunity, host genetic susceptibility, and viral genomic variation—may further shape the pathophysiology during secondary infections [8]. This immunopathologic complexity is particularly consequential in hyperendemic regions where all four DENV serotypes co-circulate, and it demands that effective vaccines induce balanced, durable immunity against all serotypes.

Multitypic dengue vaccine development has proven challenging [9,10]. The first licensed dengue vaccine - tetravalent dengue vaccine, CYD-TDV (Dengvaxia), demonstrated variable efficacy by serotype and increased the risk of severe disease in seronegative individuals [11]. More recent vaccine candidates have shown promise but continue to display uneven performance across serotypes [12]. These limitations underscore the importance of understanding local serotype distribution and transmission dynamics to guide vaccine design, evaluation, and implementation. With another live-attenuated, tetravalent dengue vaccine, TAK-003 (Qdenga), now approved for use in multiple countries, including Thailand, and is subject to ongoing post-marketing surveillance, alongside other dengue vaccines in development [13], country-specific epidemiological data are increasingly critical for informing vaccination strategies [14]. This presents the need to understand dengue epidemiology in settings with long-standing multi-serotype circulation.

Thailand represents a valuable case study for understanding dengue dynamics, as the country is hyperendemic with long-standing circulation of all four serotypes [15]. A systematic review of Ministry of Public Health (MoPH) data from 2004-2010 reported broadly consistent serotype patterns nationwide, with DENV-1 and DENV-2 predominating, DENV-3 circulated at lower but persistent levels, and DENV-4 exhibiting greater temporal and spatial variability [16]. Historically, many regions worldwide reported only one or two circulating serotypes; however, over subsequent decades there has been a steady expansion in the number of co-circulating serotypes across the Americas and Caribbean [17]. This trend suggests that Thailand's long-standing hyperendemic profile may foreshadow the epidemiologic state that other regions may be approaching. Understanding the serotype-specific epidemiology in hyperendemic contexts is therefore essential for anticipating disease burden and guiding future surveillance and vaccine strategies.

As dengue epidemiology continues to evolve regionally, serotypes that are currently underrepresented in certain settings may become more relevant as multi-serotype co-circulation becomes established. In regions experiencing sequential serotype introduction, apparent differences in serotype importance may reflect transitional epidemiologic phases rather than intrinsic differences in public health relevance. These dynamics may have contributed to perceptions that certain serotypes, such as DENV-2, are more clinically consequential, whereas others, such as DENV-4, cause milder clinical presentations in some settings [18,19]. However, such impressions are largely derived from regionally and temporally constrained datasets and may reflect differences in transmission intensity, population immunity, and surveillance practices. Evidence from long-standing hyperendemic settings supports this interpretation. For example, a meta-analysis reported differential associations between serotypes, infection sequence, and severe outcomes, while also noting substantial heterogeneity, limited representation of certain serotypes—particularly DENV-4—and potential publication bias [20]. Accordingly, low observed prevalence should not be equated with low public health importance.

To address gaps in long-term serotype-specific epidemiology, we present nearly two decades of passive surveillance data from Kamphaeng Phet, Thailand, spanning 2004–2022. The long duration of surveillance provides a unique opportunity to examine sustained serotype co-circulation, shifts in dominant serotypes, and the contribution of serotypes that may appear infrequent in shorter-term or regionally limited datasets. Because passive surveillance preferentially captures symptomatic infections that present to clinical care, the data reflects a direct measure of serotype-specific contributions to clinically apparent dengue burden over time. By characterizing serotype-specific epidemiologic patterns across nearly two decades, our study offers insight into how all four DENV serotypes contribute to dengue disease burden and provides context relevant to regions currently undergoing epidemiologic transition toward hyperendemicity.

## Methods

### Ethics statement

The surveillance study was approved by Thailand Ministry of Public Health (MoPH) Ethical Research Committee (#10/2558), Siriraj Ethics Committee on Research involving Human Subjects (#593/2557), Institutional Review Board for the Protection of Human Subjects State University of New York Upstate Medical University (#2014-5), and Walter Reed

Army Institute of Research Institutional Review Board (#2819). Written informed consent was obtained from all adult participants, and from a parent or legal guardian for all pediatric participants.

Data were obtained from passive surveillance of patients presenting with acute febrile illness. All data received and analyzed in this study were de-identified.

**Passive surveillance:.** Data were obtained from Kamphaeng Phet Hospital, a 410-bed provincial hospital in the lower northern region of Thailand. Samples were collected from patients presenting to the hospital with suspected dengue infection according to WHO case definitions in use at that time [5,21]. Majority of patients were recruited from inpatient settings, with some recruited from outpatient department. Demographic information, including age, sex, and district of residence, was recorded at the first visit. Blood samples were collected on presentation and at time of discharge for the laboratory testing below.

**PCR detection and serology:.** All samples were tested using an RT-PCR assay to confirm dengue infection, followed by nested PCR on positive samples to determine the infecting serotype [22]. Co-infection was defined as the detection of two serotypes in the same sample. Dengue-specific IgM and IgG antibodies were measured by ELISA, and paired samples collected at least five days apart were used to classify infections as acute primary or secondary infection based on established serologic criteria and expert interpretation [23]. Cases without adequate follow-up specimens were considered non-diagnostic, and those lacking serologic evidence of recent flavivirus infection by ELISA were categorized as "ELISA negative" in the figures below.

**Statistical analysis:.** Descriptive analyses were performed using R software (version 4.4.2). Case characteristics were summarized as percentages for categorical variables and as medians with interquartile ranges for continuous variables, as appropriate. Differences in the proportion of infection types among the four dengue serotypes were first assessed using the chi-square test for categorical variables or the Kruskal-Wallis rank sum test for continuous variables. Follow-up analyses were conducted using multivariable logistic regression models, with DENV-1 used as the reference serotype. To account for non-linear age effects in modeling dengue epidemiological patterns, we applied generalized additive models (GAMs) with binomial family, comparing each serotype (DENV-2, DENV-3, and DENV-4) individually against DENV-1. Models included a smooth term for age, fixed effects for infection type, and random effects for calendar year to account for year-to-year variation in serotype distribution. P-values ≤ 0.05 were considered statistically significant.

## Results

**Study population and surveillance overview.** During the passive surveillance period from 2004 to 2022, 16,108 individuals were tested for dengue, among whom 6,841 (42.5%) PCR-confirmed dengue cases were identified. One case representing a co-infection with DENV-1 and DENV-3 was excluded from subsequent analysis. Males and females were represented in approximately equal proportions. The overall age distribution was left-skewed, with the highest burden observed among older children, adolescents, and young adults aged 10–21 years, who accounted for approximately half of all cases (Table 1). Age distribution varied by serotype, as described below.

**Temporal patterns and serotype distribution.** Dengue infections exhibited marked seasonality, peaking during Thailand's rainy season (May-October), with the highest cumulative admitted case counts occurring in July (S1 Fig). All four dengue serotypes circulated throughout the nearly two-decade study period, with substantial year-to-year variation in the predominant serotype. Overall, DENV-1 through DENV-4 accounted for 32.8%, 25.9%, 24.8%, and 16.5% of cases, respectively. Fig 1 depicts annual dengue case counts alongside the predominant serotype of each year.

**Clinical burden and serotype-specific differences:** Within this passive surveillance system, which predominantly captured patients from inpatient department, nearly all detected dengue cases across serotypes were hospitalized (96.0%-97.6%). Among the 6,614 hospitalized dengue cases, DENV-1 through DENV-4 accounted for 32.7%, 25.7%, 24.9%, and 16.7% of cases, respectively.

**Table 1. Distribution of demographic among all dengue cases detected from passive surveillance by DENV serotype.**

| | Overall N = 6,840[a] | Outpatient N = 226[a] | Inpatient N = 6,614[a] |
|---|---|---|---|
| **Age** | 15.0 (10.0, 21.0) | 13.0 (9.0, 20.0) | 15.0 (10.0, 21.0) |
| **Sex** | | | |
| Female | 3,424 (50.1%) | 105 (46.5%) | 3,319 (50.2%) |
| Male | 3,416 (49.9%) | 121 (53.5%) | 3,295 (49.8%) |
| **Serotype** | | | |
| DENV-1 | 2,242 (32.8%) | 81 (35.8%) | 2,161 (32.7%) |
| DENV-2 | 1,769 (25.9%) | 70 (31.0%) | 1,699 (25.7%) |
| DENV-3 | 1,697 (24.8%) | 48 (21.2%) | 1,649 (24.9%) |
| DENV-4 | 1,132 (16.5%) | 27 (11.9%) | 1,105 (16.7%) |

[a]Median (Q1, Q3); n (%).

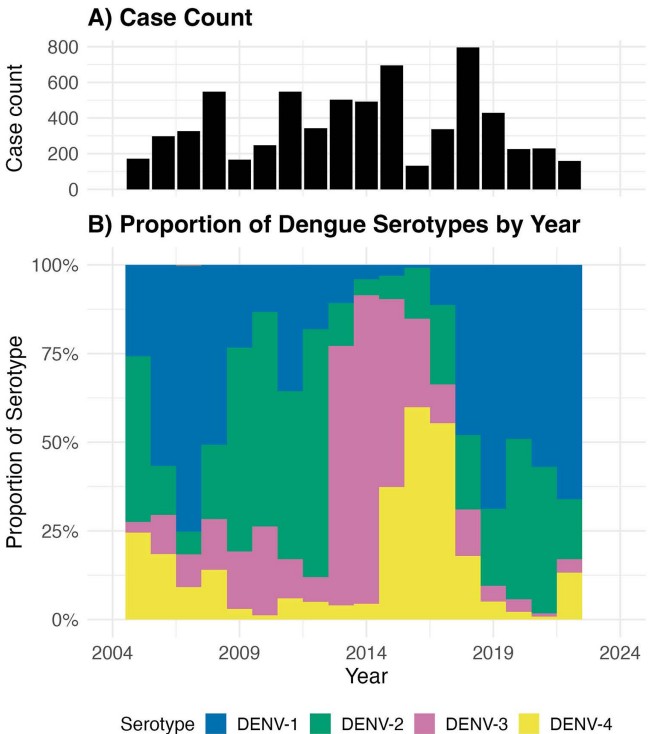

**Fig 1. Temporal trends of DENV serotypes in Kamphaeng Phet, Thailand, showing co-circulation of all four serotypes with alternating dominance by year. A)** Total case count by year and **B)** proportion serotypes detected among all cases detected via passive surveillance.

Among hospitalized cases with determined infection status, DENV-1 (54.0%) and DENV-3 (35.0%) accounted for the largest proportions of primary infections. In contrast, serotype distribution among hospitalized secondary infections was more evenly distributed, with DENV-1 through DENV-4 accounting for 29.4%, 25.0%, 26.5%, and 19.0% of cases, respectively. Serotype distribution by admission status is shown in Fig 2.

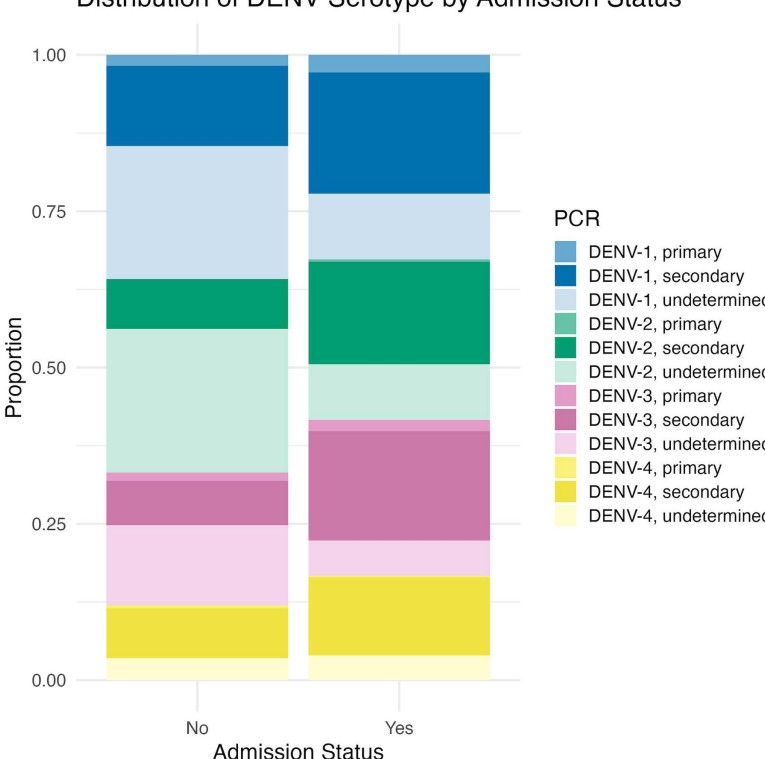

**Fig 2. Proportions of DENV serotypes and infection types stratified by hospitalization status.** PCR-confirmed infections were classified by paired serology into primary or secondary. Undetermined serology status included both non-diagnostic from inadequate follow-up and ELISA-negative results.

Across serotypes, secondary infections comprised a greater proportion of hospitalized cases, accounting for 59.3%, 64.0%, 70.1%, and 75.0% of hospitalized cases for DENV-1 through DENV-4, respectively. Confirmed primary infections represented a smaller fraction of hospitalized cases for each serotype (8.7%, 1.5%, 7.3%, and 1.2%, respectively). Table 2 presents serotype-specific distributions restricted to cases with determined infection status (primary or secondary). **Association between infection sequence and serotype.** All four dengue serotypes were detected in both primary and secondary infections; however, secondary infections predominated across all serotypes, ranging from 58.5% in DENV-1 to

**Table 2. Serotype distribution among hospitalized dengue cases with determined infection sequence status.**

| Serotype | Primary N = 226[a] | Secondary N = 6,614[a] |
|---|---|---|
| DENV-1 | 191 (12.7%) | 1,311 (87.3%) |
| DENV-2 | 25 (2.2%) | 1,106 (97.8%) |
| DENV-3 | 124 (9.6%) | 1,172 (90.4%) |
| DENV-4 | 14 (1.6%) | 847 (98.4%) |

[a]n (%).

Percentages are calculated among cases classified as primary or secondary infection. Cases with non-diagnostic serology or no serologic evidence of recent infection were excluded from these denominators.

74.8% in DENV-4 (Fig 3). While descriptive analysis suggested a higher proportion of secondary infection for DENV-3 and DENV-4, multivariable regression adjusted for age and calendar year—restricted to cases with known infection sequence status and excluding non-diagnostic and no-evidence categories—demonstrated that only DENV-2 and DENV-4 were associated with significantly greater odds of secondary infection compared with DENV-1 (odds ratio [OR]: 4.94 and 5.24, respectively; Table 3). DENV-3 did not differ significantly from DENV-1.

Age demonstrated a non-linear association with secondary infection status (estimated degrees of freedom [edf] = 2.13, p<0.001), with the probability of secondary infection increasing more steeply at younger ages and stabilizing through adulthood (Fig 4). Confidence intervals widened at older ages, reflecting the smaller number of infections observed in these age groups.

**Age distribution and serotype-specific associations.** Age distributions differed by dengue serotype. The median age and interquartile range were higher for DENV-2, DENV-3, and DENV-4 compared with DENV-1. This pattern is illustrated in Fig 5, which shows increasing proportions of DENV-2 and DENV-4 among older age categories.

In multivariable generalized additive models adjusting for infection sequence and calendar year, age demonstrated a significant non-linear association with the odds of DENV-2 versus DENV-1 infection (edf = 2.43, p<0.001). Similarly, the odds of DENV-4 infection increased non-linearly with age (edf = 1.59, p<0.001). Secondary infection status was independently associated with higher odds of DENV-2 (OR: 4.65, 95% CI: 2.95-7.35) and DENV-4 (OR: 5.58, 95% CI: 2.81-11.1) relative to DENV-1 (S1 Table).

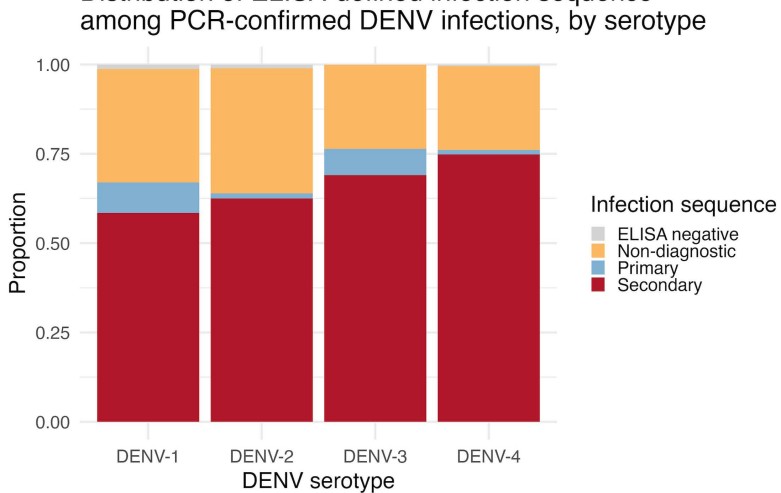

**Fig 3. Distribution of ELISA-defined infection sequence among PCR-confirmed DENV infections.** Higher proportion of secondary infections noted among DENV-3 and DENV-4 infections. Higher proportions of primary infection noted among DENV-1 and DENV-3 infections.

**Table 3. Odds ratios for secondary infection among those with clinically-apparent disease by dengue serotype from generalized additive model.**

| Predictor | Category | OR | 95% CI | P-value |
|---|---|---|---|---|
| Serotype | DENV-2 vs DENV-1 | 4.94 | 3.14-7.76 | <0.001 |
| | DENV-3 vs DENV-1 | 1.01 | 0.70-1.46 | 0.94 |
| | DENV-4 vs DENV-1 | 5.24 | 2.88-9.52 | <0.001 |

Note: DENV-1 serves as the reference serotype (OR = 1.0). Models adjusted for age using smooth terms and included calendar year as random effects. OR, odds ratio; CI, confidence interval.

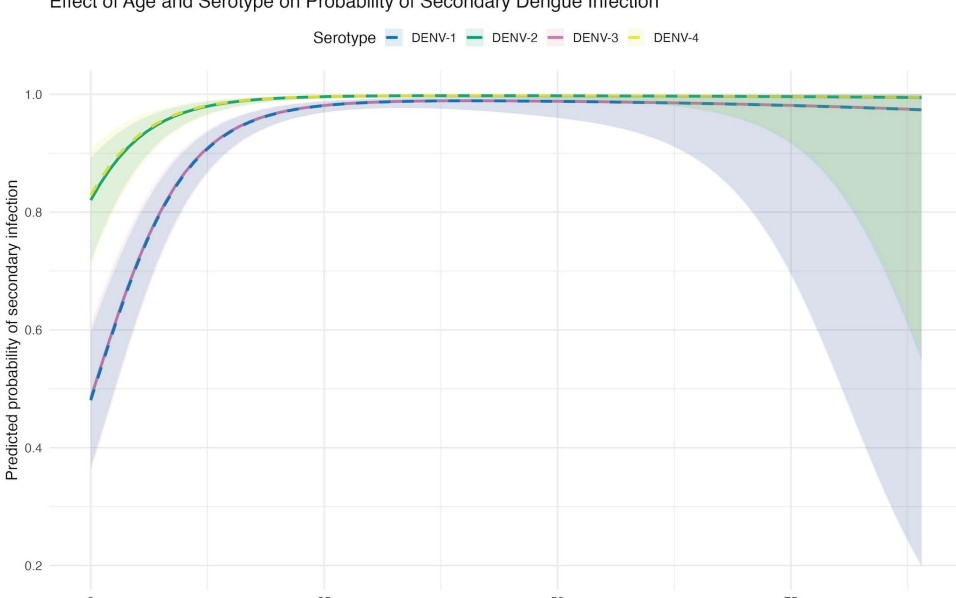

**Fig 4. Predicted probability of secondary infection by age and serotype.** Predicted probabilities from generalized additive models marginalized over year-specific outbreak effects Models showing the non-linear association between age and probability of secondary infection for each dengue serotype. Lines represent DENV-1 through DENV-4 with 95% confidence intervals (shaded regions).

In contrast, DENV-3 demonstrated an age association (edf = 1.18, p < 0.001) but no independent association with secondary infection status (p = 0.83), consistent with findings from the prior model. Significant heterogeneity across calendar years was observed, consistent with temporal variation in serotype-specific circulation. Fig 6 shows the predicted probability of each serotype by age, stratified by infection sequence.

## Discussion

Using nearly two decades of passive surveillance data, we demonstrate that all four DENV serotypes contributed meaningfully to the clinical burden of dengue illness in Thailand. Although the overall burden was shared across serotypes, their relative contributions varied by age distribution and infection sequence, reflecting heterogeneity in epidemiologic patterns. These findings have important implications for dengue surveillance, vaccine implementation, and disease control strategies in hyperendemic settings.

Throughout the surveillance period, all four DENV serotypes co-circulated in Kamphaeng Phet, with periodic shifts in the predominant serotype. Importantly, clinically apparent dengue cases presenting to the healthcare system were distributed across all four serotypes, challenging the notion that any serotype can be considered less clinically relevant. Although each serotype was detected annually, oscillations in outbreak dominance were observed, underscoring the importance of sustained, long-term surveillance. Shorter observation periods or gaps in surveillance could easily miss circulation of less dominant serotypes and lead to the false impression that certain serotypes are absent or epidemiologically unimportant.

The predominance of secondary infections among hospitalized cases across all serotypes supports the established role of sequential heterotypic infections in dengue pathogenesis. This finding is consistent with prior studies linking secondary infection to increased risk for severe dengue [19] and with the immunological evidence implicating cross-reactive immune

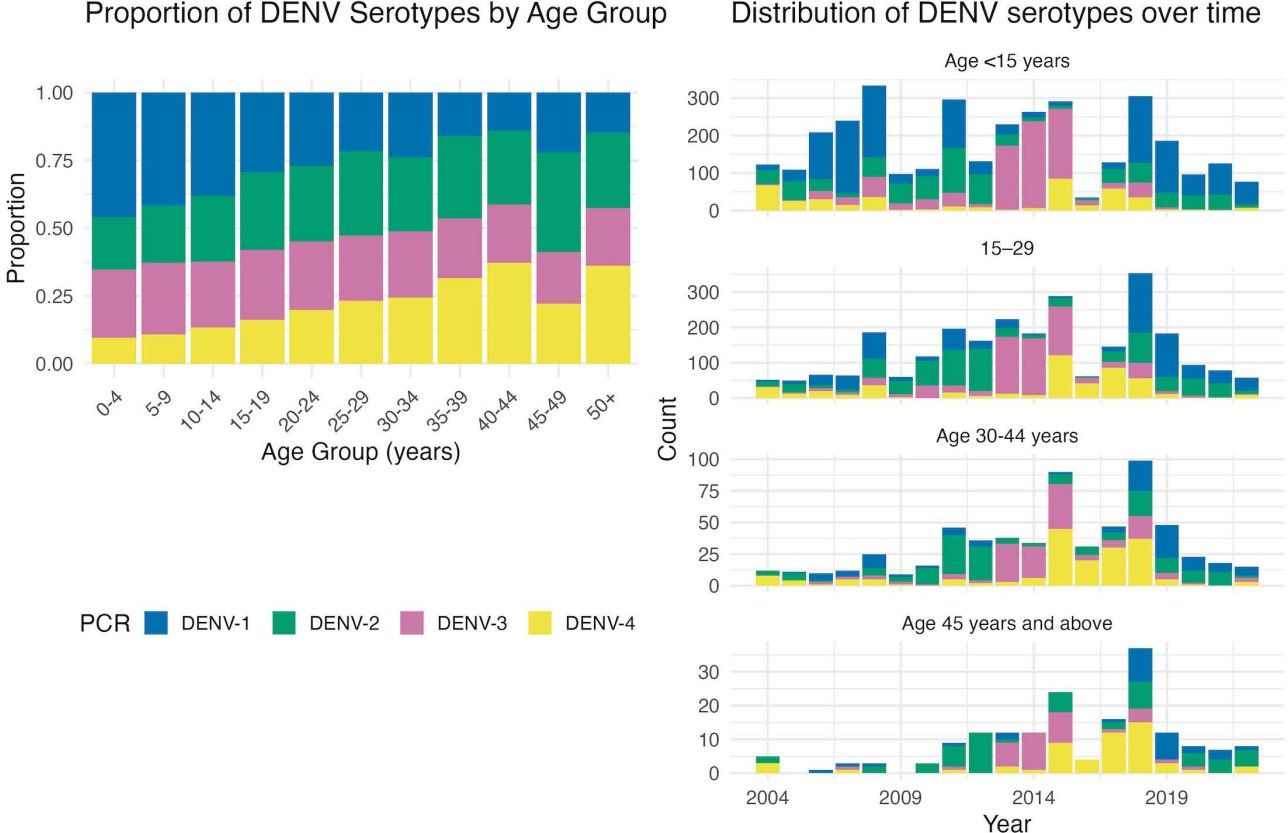

**Fig 5. Distribution of DENV serotypes over time, stratified by age group. (A)** Demonstrates that all four DENV serotypes affect all age groups. An increasing proportion of DENV-4 infections observed in older age groups. **(B)** Highlights a higher burden of infection among younger individuals.

responses and antibody-dependent enhancement in severe outcomes [7]. While our study was not designed to assess severity outcomes directly, the higher representation of secondary infections among clinically apparent cases reinforces the importance of infection history in shaping disease presentation.

We further observed that, although all serotypes contributed to clinically apparent dengue, DENV-2 and DENV-4 were more frequently detected in older individuals and were more strongly associated with secondary infections, even after accounting for year-to-year variation in serotype predominance. In this highly endemic setting, age and infection sequence were strongly correlated—primary infections were rarely observed beyond early adulthood—precluding formal evaluation of interaction effects. Nonetheless, the persistence of these associates suggests that age- and immunity-dependent factors may influence the clinical expression of different serotypes.

The association between DENV-2 and DENV-4 and older age groups warrants cautious interpretation. One potential explanation involves cohort effects or population aging; however, the observed associations persisted after adjusting for calendar year. Alternative hypotheses include viral evolution. DENV-4 has been reported to exhibit a relatively higher evolutionary rate compared with other serotypes, which could facilitate emergence of new DENV-4 lineages capable of escaping preexisting immunity [24]. Prior studies have documented genotype replacement events associated with dengue epidemics in Puerto Rico (1994), Nicaragua (2006), and South India (2017), including clade emergence involving DENV-4 [25,26]. Another possibility is differential durability of serotype-specific immunity, which could leave older individuals more susceptible to certain serotypes over time. Although speculative, these observations raise important questions

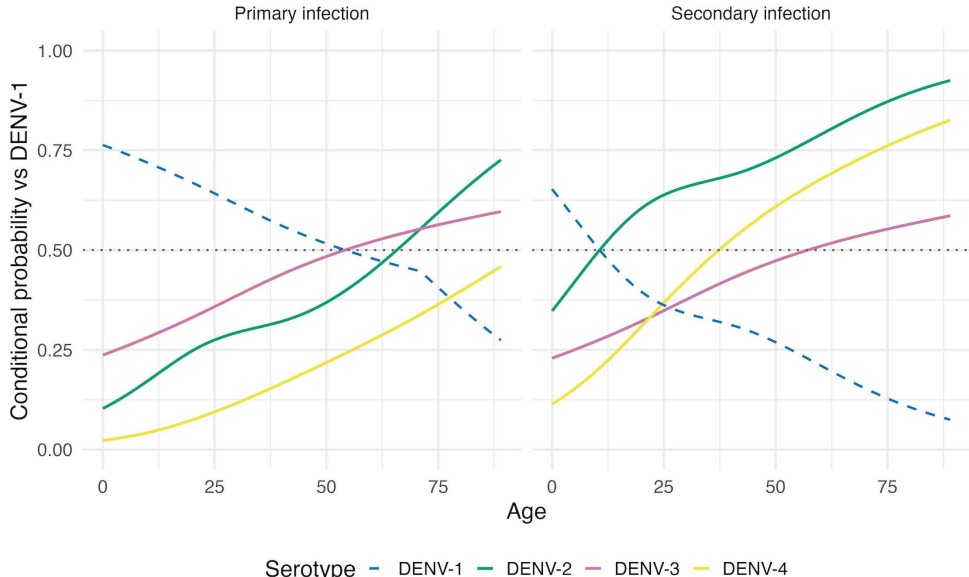

**Fig 6. Predicted probability of DENV-2, DENV-3, and DENV-4 infection by age compared to DENV-1.** Predicted probabilities from generalized additive models stratified by infection type marginalized over year-specific outbreak effects: primary infection (left panel) and secondary infection (right panel). Confidence intervals for predicted probabilities are not shown due to wide uncertainty after marginalization over calendar year. DENV-1 probability (dotted line) is approximated as 1 minus the maximum predicted probability among the other three serotypes at each age and shown for reference only.

about whether age-related differences in serotype distribution translate into variation in clinical risk, particularly in aging populations.

These findings have direct relevance for dengue vaccine development and evaluation. Effective vaccines must elicit balanced and durable immunity to all four serotypes, particularly given that vaccine efficacy has varied by serotype in clinical trials. For example, TAK-003, which is now approved for use in Thailand, demonstrated no efficacy against DENV-3 among dengue-naïve participants and inconclusive efficacy against DENV-4 in phase 3 trials [27]. Although overall efficacy was reported across serotypes among baseline seropositive participants, this group was heterogeneous and may not fully capture risks associated with secondary infections. Moreover, the trial population was limited to children and adolescents, leaving uncertainty regarding vaccine performance in older age groups. The dynamic nature of serotype predominance further emphasizes the need for ongoing serotype-specific vaccine effectiveness assessments and continued surveillance to identify potential gaps in protection.

Several limitations should be considered. Restriction to PCR-confirmed cases limited detection to the sensitivity of the assay and to the timing of sample collection, which often occurred at first clinical presentation. A substantial proportion of cases were classified as non-diagnostic for infection sequence due to incomplete follow-up sampling, likely related to early discharge among less severe cases. Although these cases were retained for analyses of serotype distribution, missing infection sequence data may bias estimates if primary infections were disproportionately underrepresented. We anticipate, however, that such bias would be relatively consistent across serotypes.

As with all passive surveillance systems, asymptomatic and mild infections that did not present for clinical care were not captured. Additionally, Kamphaeng Phet Hospital serves as a tertiary referral center, which may enrich the cohort for more clinically significant disease. Accordingly, beyond descriptive admission proportions, we did not attempt to infer

serotype-specific differences in disease severity. Despite these limitations, the use of passive surveillance provides a clear view of the serotypes contributing to clinically relevant dengue burden, which was the primary objective of this study. Integration of active surveillance in future work would allow a more complete characterization of the full spectrum of serotype-specific dengue infection, from asymptomatic infection to severe disease.

In summary, our findings demonstrate that all four DENV serotypes contribute substantially to clinically apparent dengue illness in a hyperendemic setting, with distinct age- and immunity-dependent epidemiologic patterns. Among clinically apparent cases, DENV-2 and DENV-4 infections were more likely to represent secondary infections compared with DENV-1. In contrast, DENV-1 and DENV-3 accounted for the largest proportions of clinically apparent primary infections. At the population level, overlooking any serotype risks undermining vaccine effectiveness and compromising broader control strategies. Together, these results reinforce that context matters in interpreting serotype-specific dengue burden and highlight the importance of comprehensive surveillance and truly tetravalent vaccine approaches.

## Supporting information

**S1 Table. Generalized additive model results for serotype-specific associations.** Models compare DENV-2, DENV-3, and DENV-4 against DENV-1 (reference), adjusted for age (smooth term), secondary infection status, and calendar year (random effect). Odds ratios (OR) with 95% confidence intervals (CI) are shown for parametric terms. Effective degrees of freedom (edf) and p-values are shown for smooth terms.
(DOCX)

**S1 Fig. Monthly distribution of dengue cases from 2004-2022 passive surveillance data.** Infections peaked during the rainy season in Thailand (May-October).
(TIFF)

## Acknowledgments

We acknowledge the contributions and support from Kamphaeng Phet hospitals for providing access to the data used in this analysis. Their efforts in study design, data collection, and laboratory testing were essential for this work.

Material has been reviewed by the Walter Reed Army Institute of Research. There is no objection to its presentation and/or publication. The opinions or assertions contained herein are the private views of the author, and are not to be construed as official, or as reflecting true views of the Department of the Army or the Department of Defense. The investigators have adhered to the policies for protection of human subjects as prescribed in AR 70–25.

## Author contributions

**Conceptualization:** Ruchira R. Khosavanna, Sopon Iamsirithaworn, Albert I. Ko, Kathryn B. Anderson, Darunee Buddhari.

**Data curation:** Ruchira R. Khosavanna.

**Formal analysis:** Ruchira R. Khosavanna.

**Funding acquisition:** Kathryn B. Anderson, Darunee Buddhari.

**Investigation:** Soontorn Pinpaiboon, Kanyarat Phutthasophit, Taweewun Hunsawong, Darunee Buddhari.

**Methodology:** Sopon Iamsirithaworn, Albert I. Ko, Kathryn B. Anderson, Darunee Buddhari.

**Project administration:** Soontorn Pinpaiboon, Kanyarat Phutthasophit, Taweewun Hunsawong, Darunee Buddhari.

**Resources:** Sopon Iamsirithaworn, Soontorn Pinpaiboon, Kanyarat Phutthasophit, Taweewun Hunsawong, Kathryn B. Anderson, Darunee Buddhari.

**Supervision:** Albert I. Ko, Kathryn B. Anderson, Darunee Buddhari.

**Visualization:** Ruchira R. Khosavanna.

**Writing – original draft:** Ruchira R. Khosavanna.

**Writing – review & editing:** Ruchira R. Khosavanna, Kathryn B. Anderson, Darunee Buddhari.

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
