## [Decision Letter · Decision Letter 0]

30 Apr 2026

Dear Dr. Khosavanna,

We are pleased to inform you that your manuscript 'The need for balanced dengue vaccine protection: Insights from Thai surveillance data on four serotypes' has been provisionally accepted for publication in PLOS Neglected Tropical Diseases.

Best regards,

Jonas Klingström

Academic Editor

Michael Holbrook

Section Editor

Shaden Kamhawi

co-Editor-in-Chief

Paul Brindley

co-Editor-in-Chief

Reviewer's Responses to Questions

**Key Review Criteria Required for Acceptance?**

**Methods**

-Are the objectives of the study clearly articulated with a clear testable hypothesis stated?

-Is the study design appropriate to address the stated objectives?

-Is the population clearly described and appropriate for the hypothesis being tested?

-Is the sample size sufficient to ensure adequate power to address the hypothesis being tested?

-Were correct statistical analysis used to support conclusions?

-Are there concerns about ethical or regulatory requirements being met?

Reviewer #1: see comments made in the final box.

Reviewer #2: -Are the objectives of the study clearly articulated with a clear testable hypothesis stated?  Yes

-Is the study design appropriate to address the stated objectives?  Yes

-Is the population clearly described and appropriate for the hypothesis being tested?  Yes

-Is the sample size sufficient to ensure adequate power to address the hypothesis being tested?  Yes

-Were correct statistical analysis used to support conclusions?  yes

-Are there concerns about ethical or regulatory requirements being met?  Yes

**Results**

-Does the analysis presented match the analysis plan?

-Are the results clearly and completely presented?

-Are the figures (Tables, Images) of sufficient quality for clarity?

Reviewer #1: see remarks in the box below

Reviewer #2: -Does the analysis presented match the analysis plan?  Yes

-Are the results clearly and completely presented?  yes

-Are the figures (Tables, Images) of sufficient quality for clarity?  yes

**Conclusions**

-Are the conclusions supported by the data presented?

-Are the limitations of analysis clearly described?

-Do the authors discuss how these data can be helpful to advance our understanding of the topic under study?

-Is public health relevance addressed?

Reviewer #1: This is the textbook description of what happens in a population of a tropical country that supports the transmission of all four types of dengue viruses. It describes the clinical outcomes of dengue infections that actually happened in populations where all 4 dengue viruses were endemically transmitted. As described in the text, disease burdens encumbered by each of the four dengue viruses are remarkably similar. Both primary and secondary heterotypic dengue infections by all four dengue viruses occur and cause overt symptoms. For all four types, second heterotypic dengue infections consistently cause a higher fraction of severe disease than do primary dengue infections. For a long time to come, this will be the encyclopedic reference used to justify the need for and value of tetravalent dengue vaccines. The reviewer shares with the authors a silent prayer that an efficient and fully competent all-purpose dengue vaccine will someday emerge. Products on the market or soon to finish efficacy trials might be improved. The efficacy of administering two different dengue vaccines should be fully studied. They might work!

Reviewer #2: -Are the conclusions supported by the data presented?  Yes

-Are the limitations of analysis clearly described?  yes

-Do the authors discuss how these data can be helpful to advance our understanding of the topic under study?  yes

-Is public health relevance addressed?  yes

**Editorial and Data Presentation Modifications?**

Reviewer #1: This is an important contribution to the dengue literature and necessary to justify the effort and expense of producing tetravalent dengue vaccines

Reviewer #2: Accept

**Summary and General Comments**

Reviewer #1: (No Response)

Reviewer #2: Summary of research

The authors investigated data from passive surveillance collected from Kamphaeng Phet, Thailand (2004–2022) and characterized longterm DENV serotype circulation in relation to clinical burden in a endemic setting. The results showed sustained co-circulation of all four serotypes for almost two decades including a shift in the dominant serotype with periodic shifts in dominant serotype. They included analysed 6,840 PCR-confirmed dengue cases where DENV-1 was predominant, furthermore, they show that DENV-2 and DENV-4 infections were more represented in secondary infections.

Comments

The manuscript is well written and easy to understand. It strongly contributes to the scientific community within the field of dengue, ADE, secondary infections and surveillance.

Major conserns

None

Minor concerns

None

Overall impression

The study design, data from passive surveillance, is elegant and nicely designed to extract the data. The data provided in this study contribute to the understanding of disease burden of dengue in Thailand, and is also of importance for other endemic countries. As there are new dengue vaccines upcoming, the data is of importance for vaccine recommendations in endemic countries, but also in the area of travel medicine for non-endemic countries. The manuscript is easy to follow and well written. Tables and figures are relevant, I do not see additional data needed to confirm the outlined question.

PLOS authors have the option to publish the peer review history of their article (what does this mean?). If published, this will include your full peer review and any attached files.

Reviewer #1: **Yes:** Scott B Halstead

Reviewer #2: **Yes:** Kim Blom

---

## [Editor Report · Acceptance letter]

Dear Dr. Khosavanna,

We are delighted to inform you that your manuscript, "The need for balanced dengue vaccine protection: Insights from Thai surveillance data on four serotypes," has been formally accepted for publication in PLOS Neglected Tropical Diseases.

Best regards,

Shaden Kamhawi

co-Editor-in-Chief

Paul Brindley

co-Editor-in-Chief
